# Towards AGI: Future Directions, Theoretical Limits, and Deployment Challenges of Autonomous AI Agents

Lakshay Dagar
Department of Computer Science and Engineering
Chitkara University
lakshay1841.be22@chitkara.edu.in

Shresth kumar
Department of Computer Science and Engineering
Chitkara University
Shresth2340.be22@chitkara.edu.in

Aryan
Department of Computer Science and Engineering
Chitkara University
Aryan0175.be22@chitkara.edu.in

Rohit Thakur
Department of Computer Science and Engineering
Chitkara University
Rohit2180.be22@chitkara.edu.in

Rajat Takkar
Department of Computer Science and Engineering
Chitkara University
Rajat.takkar@chitkara.edu.in

**Abstract**

AGI refers to the development of artificial systems that can perform a wide range of cognitive tasks with flexibility comparable to human intelligence. While modern artificial intelligence systems have achieved significant success in specialized domains such as image recognition, language generation, and decision support, they remain fundamentally limited to narrow problem spaces. These systems typically perform well only within the scope of the tasks they were trained for and struggle when presented with unfamiliar situations.

Recent advances in machine learning, particularly the development of LLMs and transformer-based neural networks, have significantly improved the capabilities of AI systems. These models are capable of generating human-like text, assisting with programming, summarizing information, and answering complex questions. However, these capabilities alone are not sufficient for achieving general intelligence.

One emerging approach to expanding the capabilities of AI systems involves the development of Autonomous AI Agents. Autonomous agents extend language models by integrating planning mechanisms, memory storage systems, and external tools such as search engines and databases. These systems can break down complex goals into smaller tasks and execute them sequentially.

This paper presents a practical exploration of Autonomous AI Agents through the design and implementation of a prototype research assistant agent. The proposed system architecture integrates a language model with a task planner, vector-based memory database, and tool interface for external data retrieval. Experimental evaluation demonstrates that the prototype agent can successfully complete structured research tasks such as document summarization and information retrieval.

The findings highlight both the potential and limitations of current autonomous agent systems and provide insights into future research directions toward more adaptable and intelligent AI systems.

**Keywords**— Artificial General Intelligence (AGI), Large Language Models (LLMs), Natural Language Processing (NLP), Intelligent Systems(IS),Artificial Intelligence(AI).

# 1. INTRODUCTION

Artificial intelligence has evolved through several distinct paradigms since the field was formally introduced in the mid-twentieth century. Early AI research focused primarily on symbolic reasoning systems that relied on formal logic and rule-based knowledge representations. Researchers believed that intelligent behavior could be achieved by encoding expert knowledge into large collections of logical rules [1].

While symbolic systems were capable of solving structured problems such as theorem proving and constraint satisfaction, they struggled when applied to complex real-world environments. These systems lacked the ability to learn from data and required extensive manual engineering to maintain.

During the 1990s and early 2000s, machine learning approaches began to dominate AI research. Instead of relying on manually designed rules, machine learning algorithms learn statistical patterns directly from data. Methods such as decision trees, support vector machines, and probabilistic graphical models were widely adopted in applications such as pattern recognition and predictive analytics [1].

The emergence of deep learning in the 2010s significantly accelerated the capabilities of AI systems. Deep neural networks with multiple layers enabled models to learn hierarchical feature representations from complex datasets. This advancement allowed AI systems to achieve breakthrough performance in tasks such as computer vision, speech recognition, and NLP [2].

A particularly important milestone occurred in 2017 with the introduction of the transformer architecture, which uses attention mechanisms to model relationships between elements in sequential data. Transformer-based models enable the training of LLMs on massive text corpora, dramatically improving performance in language understanding and generation tasks [3],[4].

Despite these advancements, most modern AI systems remain specialized tools rather than truly general intelligence systems. Humans are capable of transferring knowledge between domains, adapting to new environments, and solving unfamiliar problems using reasoning and experience. Replicating these capabilities remains one of the central challenges of AI research [5].

One promising approach involves the development of Autonomous AI Agents. Unlike traditional models that respond to isolated prompts, autonomous agents pursue long-term goals by performing sequences of reasoning steps, interacting with external tools, and storing intermediate knowledge in memory systems.

This paper investigates the design and implementation of such an autonomous agent architecture and evaluates its performance in research-related tasks.

# 2. BACKGROUND AND RELATED WORK

## 2.1 EVOLUTION OF ARTIFICIAL INTELLIGENCE

Artificial intelligence research has progressed through several major stages.

Symbolic AI dominated early research and relied on explicit knowledge representations and logical inference. Expert systems such as MYCIN demonstrated early success but required extensive manual knowledge engineering.

Machine learning approaches later replaced symbolic methods in many applications by enabling systems to learn patterns from data. Algorithms such as logistic regression and support vector machines became widely used.

Deep learning further expanded the capabilities of machine learning through neural network architectures capable of modeling complex nonlinear relationships [1].

The introduction of transformer models represented a major step forward in sequence modeling. The attention mechanism allows models to dynamically focus on different parts of the input sequence when generating outputs [3].

The attention mechanism used in transformer models can be represented mathematically as: [3]

$$\text{Attention}(Q, K, V) = \text{softmax}((QK^T) / \sqrt{d\_k}) V \quad (1)$$

where Q, K, and V represent the query, key, and value matrices, and $dkd\_kdk$ represents the dimensionality of the key vectors.

This mechanism allows models to capture long-range dependencies in sequential data.

## 2.2 AUTONOMOUS AI AGENTS

Autonomous AI Agents extend language models by integrating additional system components [4].

Typical agent architectures include:

• Task planning modules
• Memory storage systems
• External tool interfaces
• Multi-step reasoning mechanisms

These components enable systems to perform complex workflows that involve multiple stages of reasoning and information retrieval.

## 3. SYSTEM ARCHITECTURE

The proposed architecture consists of four major modules:

• Task Planner
• Execution Agent
• External Tool Interface
• Vector Memory Database

These components interact to form a goal-directed reasoning loop.

### 3.1 Task Planning Module

The task planner interprets the user query and decomposes it into smaller subtasks that can be executed sequentially.

Example subtasks may include:

- Searching for relevant research papers

- Extracting key findings

- Comparing multiple sources

- Generating a structured summary

## 3.2 Execution Agent

The execution agent is responsible for performing reasoning steps and interacting with external tools.

The agent uses a large language model to determine which action should be taken at each step of the reasoning process [4].

External Tool Interface

The system integrates several external tools including:

• Web search API
• Document retrieval system
• Python execution environment

These tools enable the agent to retrieve real-time information and perform computational tasks.

## 3.3 Memory System

The memory module stores retrieved documents and intermediate reasoning outputs using vector embeddings.

The similarity between two vectors is computed using cosine similarity:

Similarity(q, d) = (q · d) / (||q|| ||d||) (2)

where q represents the query embedding and d represents the document embedding.

This allows the system to retrieve semantically similar information during later reasoning steps.

## 4. ALGORITHM DESIGN

The reasoning process of the autonomous agent follows an iterative workflow.

Algorithm 1: Autonomous Research Agent

Input: User Query Q
Output: Final Generated Response R

1. Receive user query Q
2. Initialize task list T using task planner
3. Initialize memory database M

4. For each task t in T:
    Retrieve relevant context from memory M
    If required:
        call external tool (search / database / API)
    Generate intermediate result r using language model

Store result r in memory database M

5. Combine intermediate results
6. Generate final response R
7. Return R [4]

This algorithm allows the system to perform multi-step reasoning while maintaining contextual memory.

## 5. IMPLEMENTATION DETAILS

The prototype system was implemented using the following technologies:

| Component | Technology |
|---|---|
| Language Model | GPT-class LLM API |
| Framework | LangChain |
| Programming Language | Python 3.10 |
| Vector Database | FAISS |
| Embedding Model | Sentence-Transformers |
| Search Tool | SerpAPI |

The system was developed on a workstation with the following configuration:

CPU: Intel i7 processor
RAM: 16 GB
Operating System: Ubuntu Linux

The agent workflow was implemented using the LangChain agent executor, which manages the reasoning loop and tool selection process.

## 6. EXPERIMENTAL SETUP

To evaluate the effectiveness of the proposed system, a set of research-oriented tasks was constructed.

Evaluation Dataset

The dataset consisted of 25 research queries derived from academic papers and technical documentation.

Example queries included:

- Summarize key contributions of reinforcement learning research [6]

- Compare transformer models with recurrent neural networks [3]

- Explain the concept of meta-learning

Evaluation Metrics

Three metrics were used for evaluation:

1. Task Completion Rate

2. CompletionRate = SuccessfulTasks / TotalTasks (3)

Average number of reasoning iterations required.

3. Response Quality

Human evaluators rated responses on a 5-point scale based on:

- accuracy

- completeness

- clarity

## 7. EXPERIMENTAL RESULTS

| Task | Completion Rate | Avg Steps | Quality Score |
|------|-----------------|-----------|---------------|
| Literature Review | 80% | 6 | 4.3 |
| Topic Summarization | 84% | 5 | 4.4 |
| Multi-Step QA | 68% | 8 | 3.8 |
| Information Retrieval | 76% | 7 | 4.0 |

The agent performed particularly well in structured summarization tasks where relevant documents were successfully retrieved.

However, performance decreased for tasks requiring longer reasoning chains.

## 8. DISCUSSION

The experimental results demonstrate that integrating planning, memory retrieval, and tool usage can significantly improve the capabilities of language-model-based systems. The vector memory system played a critical role in maintaining contextual information during multi-step reasoning processes. By retrieving semantically related documents, the system was able to produce more consistent responses. However, several challenges were observed. When reasoning chains became too long, the probability of intermediate errors increased. Additionally, the task planner occasionally generated redundant subtasks that resulted in inefficient execution loops. Another challenge relates to memory scalability. As the memory database grows, maintaining efficient retrieval performance becomes increasingly important. Future research may explore hierarchical memory systems or knowledge graphs to address these limitations.

## 9. FUTURE WORK

Several directions may improve Autonomous AI Agents:

• Hybrid neural-symbolic architectures
• Improved long-term memory systems
• Distributed multi-agent collaboration frameworks
• Hardware accelerators for large-scale AI workloads

Advances in these areas may significantly enhance the scalability and reliability of autonomous agent systems.

## 10. CONCLUSION

Artificial intelligence has achieved remarkable progress through the development of machine learning and deep neural network technologies. Nevertheless, most current AI systems remain limited to narrow task domains.

Autonomous AI Agents represent an important architectural approach for extending the capabilities of modern AI systems. By integrating planning mechanisms, memory retrieval systems, and external tools, agent architectures enable AI systems to perform complex multi-step tasks. This paper presented the design and implementation of a prototype autonomous research assistant agent and evaluated its performance across several research-oriented tasks.

The results demonstrate the potential of agent-based architectures while also highlighting areas that require further improvement.

Continued research into reasoning algorithms, memory architectures, and multi-agent systems will be essential for advancing toward the long-term goal of AGI.

## 11. Implementation Table:

| Language Model | GPT-class LLM API |
|---|---|
| Framework | LangChain |
| Programming Language | Python 3.10 |
| Vector Database | FAISS |
| Embedding Model | Sentence-Transformers |
| Search Tool | SerpAPI |

## 12. Results Table:

| Task | Completion Rate | Avg Steps | Quality Score |
|---|---|---|---|
| Literature Review | 80% | 6 | 4.3 |
| Topic Summarization | 84% | 5 | 4.4 |
| Multi-Step QA | 68% | 8 | 3.8 |
| Information Retrieval | 76% | 7 | 4.0 |

## 13. ACKNOWLEDGMENT

The author thanks Chitkara University for providing the foundational support, hardware, and academic resources necessary to conduct this research on edge-based mental health monitoring. Special thanks to Peer for their valuable guidance and feedback during the development of the CNN pipeline, and to the faculty for their insights into ethical deployment strategies in higher education.

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
