# OpenReview forum: "Towards AGI: Future Directions, Theoretical Limits, and Deployment Challenges of Autonomous AI Agents"
_NortheastGenAI/2026/Workshop — NortheastGenAI 2026 Workshop Submission_

### Official Review · ~Badal_Nyalang1 · 2026-05-23
**No NE India connection, multiple CFP violations — Reject**

**Rating:** 2
**Confidence:** 5

**Review:**

**Relevance: None**
There is zero connection to Northeast India anywhere in this paper. Not in the abstract, not in the experiments, not in a single sentence. The acknowledgment section mentions "edge-based mental health monitoring" and a "CNN pipeline" — suggesting this may be a repurposed paper from a completely different submission. That alone is a serious concern.

**Plausibility: Weak**
25 queries evaluated by unspecified human evaluators with no IAA reported. "GPT-class LLM API" is not a reproducible specification. The implementation table is duplicated verbatim as a separate section. The acknowledgment references research that has nothing to do with this paper.

**Novelty: Weak**
A generic LangChain agent wrapper evaluated on generic CS queries. Nothing here advances the field.

**Clarity: Poor**
Structural issues throughout — duplicated tables, a mismatched acknowledgment, vague citations. No AI disclosure section anywhere.

**Verdict: Reject**
Fails G4 (no NE India focus), G7 (no grounding), and G2 (no AI disclosure). The mismatched acknowledgment raises questions about whether this was submitted to the right venue at all. This is the weakest submission in the batch by a significant margin.

*This review was generated with AI assistance and checked by the workshop chairs.*

---

### Decision · Program_Chairs · 2026-05-23

**Decision:**

Reject

**Comment:**

This paper has no connection to Northeast India. The research focus, examples, and evaluation are entirely domain-general. The paper also lacks an AI disclosure statement and contains structural inconsistencies including a mismatched acknowledgment section that references unrelated research. These issues together constitute violations of G2, G4, and G7 of the CFP.

Decision: Reject